# Preclinical Evaluation of Podoplanin-Targeted Alpha-Radioimmunotherapy with the Novel Antibody NZ-16 for Malignant Mesothelioma

**DOI:** 10.3390/cells10102503

**Published:** 2021-09-22

**Authors:** Hitomi Sudo, Atsushi B. Tsuji, Aya Sugyo, Mika K. Kaneko, Yukinari Kato, Kotaro Nagatsu, Hisashi Suzuki, Tatsuya Higashi

**Affiliations:** 1Department of Molecular Imaging and Theranostics, Institute for Quantum Medical Science (iQMS), National Institutes for Quantum and Radiological Science and Technology (QST), 4-9-1 Anagawa, Inage, Chiba 263-8555, Japan; sudo.hitomi@qst.go.jp (H.S.); sugyo.aya@qst.go.jp (A.S.); higashi.tatsuya@qst.go.jp (T.H.); 2Department of Antibody Drug Development, Tohoku University Graduate School of Medicine, 2-1 Seiryo-machi, Aoba-ku, Sendai, Miyagi 980-8575, Japan; k.mika@med.tohoku.ac.jp (M.K.K.); yukinarikato@med.tohoku.ac.jp (Y.K.); 3Department of Molecular Pharmacology, Tohoku University Graduate School of Medicine, 2-1 Seiryo-machi, Aoba-ku, Sendai, Miyagi 980-8575, Japan; 4Department of Advanced Nuclear Medicine Science, Institute for Quantum Medical Science (iQMS), National Institutes for Quantum and Radiological Science and Technology (QST), 4-9-1 Anagawa, Inage, Chiba 263-8555, Japan; nagatsu.kotaro@qst.go.jp (K.N.); suzuki.hisashi@qst.go.jp (H.S.)

**Keywords:** molecular radiotherapy, improved efficacy, tumor volume reduction, prolonged survival, actinium-225

## Abstract

The prognosis of advanced mesothelioma is poor. Podoplanin (PDPN) is highly expressed in most malignant mesothelioma. This study aimed to evaluate the potential alpha-radioimmunotherapy (RIT) with a newly developed anti-PDPN antibody, NZ-16, compared with a previous antibody, NZ-12. Methods: The in vitro properties of radiolabeled antibodies were evaluated by cell binding and competitive inhibition assays using PDPN-expressing H226 mesothelioma cells. The biodistribution of ^111^In-labeled antibodies was studied in tumor-bearing mice. The absorbed doses were estimated based on biodistribution data. Tumor volumes and body weights of mice treated with ^90^Y- and ^225^Ac-labeled NZ-16 were measured for 56 days. Histologic analysis was conducted. Results: The radiolabeled NZ-16 specifically bound to H226 cells with higher affinity than NZ-12. The biodistribution studies showed higher tumor uptake of radiolabeled NZ-16 compared with NZ-12, providing higher absorbed doses to tumors. RIT with ^225^Ac- and ^90^Y-labeled NZ-16 had a significantly higher antitumor effect than RIT with ^90^Y-labeled NZ-12. ^225^Ac-labeled NZ-16 induced a larger amount of necrotic change and showed a tendency to suppress tumor volumes and prolonged survival than ^90^Y-labeled NZ-16. There is no obvious adverse effect. Conclusions: Alpha-RIT with the newly developed NZ-16 is a promising therapeutic option for malignant mesothelioma.

## 1. Introduction

Malignant mesothelioma is an aggressive tumor that arises primarily in the pleural or peritoneal mesothelial surfaces [1]. Surgical resection is only offered to patients with early-stage disease [1,2]. Most patients reach advanced-stage disease before diagnosis, and thus the primary treatment is systemic chemotherapy [1,2]. The prognosis is poor and the median overall survival of patients who undergo chemotherapy is approximately 12 months [2]. Therefore, the development of more effective treatments for unresectable malignant mesothelioma is strongly desired.

Mesothelioma is classified into three types, epithelioid, sarcomatoid, and biphasic, based on histological characteristics [1,2]. There are several markers for the epithelioid subtype, such as calretinin, WT-1, cytokeratin 5, and ERC/mesothelin [3,4]. Those markers do not express in the sarcomatoid subtype, but podoplanin (PDPN) is overexpressed in more than 80% of all types [5,6]. PDPN is a type I transmembrane sialomucin-like glycoprotein expressed in kidney podocytes, alveolar type I cells, and lymphatic endothelial cells [7]. High expression of PDPN in tumors is associated with epithelial–mesenchymal transition, migration, invasion, and metastasis [8,9]. Several preclinical studies have demonstrated that anti-PDPN antibodies inhibit cancer metastasis [10] and cancer progression [11,12]. Therefore, PDPN is a promising therapeutic target for malignant mesothelioma.

Radioimmunotherapy (RIT) is a selective internal radiation therapy in which high-affinity antibodies against tumor-associated antigens are used to transport radionuclides to tumors [13]. In clinical practice, RIT for hematologic malignancies such as non-Hodgkin’s lymphoma utilizes anti-CD20 antibodies conjugated with β-emitters, ^90^Y or ^131^I, and the overall response rates are high, reaching 60–80%, with a complete remission rate of 15–40% [13,14]. The clinical efficacy of existing RIT for solid tumors, however, remains low, mainly due to the low radiosensitivity of solid tumors. Overcoming the radioresistance is necessary to enhance the clinical efficacy of RIT.

The clinical efficacy of α-particle emitters in the treatment of solid cancer was recently demonstrated [15]. α-Particle emitters have a greater linear energy transfer compared with β-emitters and deposit more energy into tumor cells, which results in greater DNA damage to the cells [16]. Actinium-225 is an α-particle-emitting radionuclide that generates a total of four α-particles in the decay chain [17]. The half-life of ^225^Ac is appropriate for the pharmacokinetics of antibodies. Therefore, RIT with ^225^Ac is expected to improve the therapeutic efficacy of RIT treatment for solid tumors.

A previous study reported that ^90^Y-labeled anti-PDPN antibody NZ-12 suppresses tumor growth in a mesothelioma model cell line NCI-H226 (H226); unfortunately, complete remission was not achieved [6]. To improve the therapeutic effect of RIT with an anti-PDPN antibody, we newly developed an anti-PDPN antibody, NZ-16, having a different constant region than NZ-12. NZ-16 has a higher affinity than NZ-12 for H226 mesothelioma cells and is, therefore, expected to deliver more radionuclides to the tumors. In the present study, we first compared the in vitro and in vivo properties of NZ-12 and NZ-16 radiolabeled with ^111^In. After confirming that NZ-16 has more favorable binding properties than NZ-12, the antitumor effects of ^225^Ac-labeled NZ-16 were compared with those of ^90^Y-labeled NZ-16 in an H226 mesothelioma mouse model.

## 2. Materials and Methods

### 2.1. Antibody

A rat–human chimeric anti-human PDPN antibody, NZ-12, was previously generated [18]. To generate the novel chimeric anti-human PDPN antibody NZ-16, the appropriate heavy chain variable domain of a rat NZ-1 antibody [19] and heavy chain constant domain of human IgG_1_ were subcloned into the pCAG-Neo vector (FUJIFILM Wako Pure Chemical Corporation, Osaka, Japan), and the light chain variable domain of a rat NZ-1 antibody and human lambda light chain constant domain were subcloned into pCAG-Ble vectors (FUJIFILM Wako Pure Chemical Corporation). The vectors were transfected into ExpiCHO-S cells using the ExpiCHO Expression System (Thermo Fisher Scientific Inc., Waltham, MA, USA). NZ-16 was purified using Protein G-Sepharose (GE Healthcare BioSciences, Chicago, PA, USA).

### 2.2. Cell Culture

Mesothelioma cell line NCI-H226 (H226, CRL-5826) was obtained from ATCC (Manassas, VA, USA). The cells were cultured in RPMI-1640 (FUJIFILM Wako Pure Chemical Corporation) containing 10% fetal bovine serum (Thermo Fisher Scientific Inc.) in 5% CO_2_ at 37 °C.

### 2.3. Radiolabeling of Antibodies

For radiolabeling of radiometals ^111^In, ^90^Y, and ^225^Ac, antibodies are necessary to be conjugated with a chelator. The present study employed *p*-SCN-Bn-DOTA (DOTA, Macrocyclics, Dallas, TX, USA). All radiolabeled antibodies were conjugated with DOTA, but DOTA was abbreviated; namely, we presented them like ^111^In-, ^90^Y-, and ^225^Ac-labeled antibodies. Antibodies were conjugated with DOTA as described previously [20]. Briefly, antibodies (5 mg/mL) were reacted with four equal molar amounts of DOTA in 50 mM borate buffer (pH8.5) for 16 h at 37 °C. The conjugation ratios of DOTA to antibodies were estimated to be approximately 2.8 each as determined by radio-thin-layer chromatography with 80% methanol. The DOTA-conjugated antibodies were purified by elution with 0.1 M acetate buffer (pH 6.0) using a Sephadex G-50 column (GE Healthcare BioSciences). ^111^InCl_3_ (Nihon Medi-Physics, Tokyo, Japan) or ^90^YCl_3_ (Perkin Elmer, Waltham, MA, USA) was incubated in 0.5 M acetate buffer (pH 6.0) for 5 min at room temperature. Each was mixed with the DOTA-antibody conjugate and incubated for 60 min at 37 °C. Radiolabeling of the antibody with ^225^Ac was conducted as previously described [20]. ^225^AcNO_3_ (Oak Ridge National Laboratory, Oak Ridge, TN, USA) dissolved in 0.2 M optima grade HCl (Thermo Fisher Scientific Inc.) was added to 2 M tetramethylammonium acetate (Tokyo Chemical Industry, Tokyo, Japan) and 150g/L L-ascorbic acid (MilliporeSigma, St. Louis, MO, USA), and the solution was incubated for 5 min at room temperature. The solution was then mixed with the DOTA-conjugated antibodies and incubated for 60 min at 37 °C. The radiolabeled antibodies were purified using an Amicon Ultra centrifugal filter (Merck Millipore, Darmstadt, Germany), and the purified antibodies were analyzed by radio-thin layer chromatography. The specific activity of ^111^In-labeled NZ-12, ^111^In-labeled NZ-16, ^90^Y-labeled NZ-16, and ^225^Ac-labeled NZ-16 was approximately 4.9 ± 2.5, 9.7 ± 4.5, 662.3 ± 151.8, and 0.6 ± 0.1 kBq/μg, respectively. The radiochemical yield was approximately 40% for ^111^In-labeled NZ-12, 50–80% for ^111^In-labeled NZ-16, 90% for ^90^Y-labeled NZ-16, and 25% for ^225^Ac-labeled NZ-16. The radiochemical purities were greater than 95% after purification.

### 2.4. Cell Binding and Competitive Inhibition Assays

For the cell binding assays, H226 cells (1.0 × 10^7^, 5.0 × 10^6^, 2.5 × 10^6^, 1.3 × 10^6^, 6.3 × 10^5^, 3.1 × 10^5^, 1.6 × 10^5^, and 7.8 × 10^4^) in phosphate-buffered saline with 1% bovine serum albumin (MilliporeSigma) were incubated with ^111^In-labeled NZ-12 or NZ-16 antibodies on ice for 60 min. After washing, cell-bound radioactivity was measured using a gamma counter (Wizard2 Automatic Gamma Counters, PerkinElmer, Waltham, MA, USA). For competitive inhibition assays, H226 cells (1.0 × 10^6^) in phosphate-buffered saline with 1% bovine serum albumin were incubated with ^111^In-labeled NZ-12 or NZ-16 in the presence of varying concentrations of intact NZ-12, intact NZ-16, DOTA-conjugated NZ-12, or DOTA-conjugated NZ-16 antibodies (0, 0.02, 0.07, 0.2, 0.7, 2.0, 6.1, 18.2, and 54.5 nmol/L) on ice for 60 min. After washing, cell-bound radioactivity was measured with a gamma counter. The dissociation constant was estimated by applying data to a one-site competitive binding model using GraphPad Prism 8 software (ver. 8.4.3, GraphPad Software, La Jolla, CA, USA).

### 2.5. Tumor Model

The animal experimental protocol was approved by the Animal Care and Use Committee of the National Institutes for Quantum and Radiological Science and Technology (13–1022, 26 May 2016), and all animal experiments were conducted according to the institutional guidelines regarding animal care and handling. H226 cells (5 × 10^6^) were subcutaneously inoculated into male nude mice (BALB/c-nu/nu, 4 weeks old, CLEA Japan, Tokyo, Japan) under isoflurane anesthesia.

### 2.6. Biodistribution of Radiolabeled Antibodies

When tumor volumes reached approximately 50 mm^3^, mice (n = 4–5/time-point), were intravenously injected with ^111^In-labeled NZ-12 (37 kBq, n = 4/time-point), ^111^In-labeled NZ-16 (37 kBq, n = 5/time-point), or ^225^Ac-labeled NZ-16 (3.7 kBq, n = 5/time-point) in a total of 40 μg of antibody adjusted by adding the corresponding unlabeled antibodies. The mice were euthanized by isoflurane inhalation at 1, 2, 4, or 7 days after injecting the ^111^In-labeled antibodies and 4 days after injecting the ^225^Ac-labeled NZ-16. Blood was obtained from the heart, and the tumor, brain, liver, spleen, intestine, kidney, and muscle were dissected and weighed. Radioactivity was measured using a gamma counter with an energy window of 150–350 keV for ^111^In and 200–300 keV for ^225^Ac. The uptake is represented as a percentage of the injected dose (radioactivity) per gram of tissue (% ID/g).

### 2.7. Dosimetry

As described previously [21], the absorbed doses of the ^90^Y- and ^225^Ac-labeled antibodies were estimated using the area under the curve based on the biodistribution data of the ^111^In-labeled antibodies and the mean energy emitted per transition of Y-90, 1.495 × 10^−13^ Gy kg (Bq s)^−1^ [22] and that of Ac-225 and all the daughter nuclei with corrections for branching, 4.6262 × 10^−12^ Gy kg (Bq s)^−1^ [22]. The absorbed dose of bone marrow was based on the blood data, considering a red-marrow-to-blood activity ratio of 0.4 [23]. Radiation weighting factors of 1 and 5 were used for Y-90 and Ac-225, respectively, as recommended by the Medical Internal Radiation Dose Committee [24]. The estimated absorbed dose is expressed as Sv when considering the radiation weighting factors.

### 2.8. Radioimmunotherapy with ^90^Y- and **^225^Ac**-Labeled Antibody

The mice were intravenously injected with intact NZ-16 (0 megabecquerel (MBq), n = 5), ^90^Y-labeled NZ-16 (3.7 MBq, n = 5), or ^225^Ac-labeled NZ-16 (11,1 and 18.5 kBq, n = 5) antibodies at a total of 40 μg of antibody adjusted by adding the corresponding unlabeled antibodies. Tumor sizes and body weights were measured at least twice a week for 8 weeks after administration. Tumor size was measured using a digital caliper, and tumor volume was calculated according to the following formula: tumor volume (mm^3^) = (length × width^2^)/2. When the tumor volume reached greater than 800 mm^3^ or body weight loss was more than 20% compared with that at day 0, the mouse was euthanized humanely by isoflurane inhalation.

### 2.9. Histologic Analysis

H226 tumors were resected from mice on days 1, 3, and 7 post injection with intact NZ-16 (0 MBq, n = 3/time-point), ^90^Y-labeled NZ-16 (3.7 MBq, n = 3/time-point), or ^225^Ac-labeled NZ-16 (18.5 kBq, n = 3/time-point). The tumors were fixed in 10% neutral-buffered formalin and embedded in paraffin. The tumor sections (1 µm thick) were deparaffinized and stained with hematoxylin and eosin. Tumor cell proliferation was evaluated by Ki-67-immunohistochemical staining with a rabbit anti-Ki-67 antibody (SP6, Abcam, Cambridge, MA, USA) and an anti-rabbit HRP/DAB Detection Kit (Abcam) according to the manufacturer’s instructions. The Ki-67 index was calculated by counting the percentage of Ki-67-positive tumor cells per >2500 tumor cells in a section with 200 × magnification (n = 3).

### 2.10. Statistical Analysis

Data are expressed as the means ± standard deviation. Statistical analysis was performed using GraphPad Prism 8 software (ver. 8.4.3). Cell binding data and tumor volume data were analyzed by two-way ANOVA. Ki-67 staining data were analyzed by one-way ANOVA with Tukey’s multiple comparison post hoc test. Uptake data of radiolabeled antibodies were analyzed by unpaired *t*-test. Log-rank tests were used to evaluate Kaplan–Meier survival curves based on a tumor volume endpoint of 300 mm^3^. *p* < 0.05 was considered statistically significant in all experiments.

## 3. Results

### 3.1. In Vitro Characterization of the Antibodies

To confirm the reactivity of NZ-16 in H226 cells, immunofluorescence staining was conducted. A strong intensity was observed on the cell membranes of H226 (Appendix A). In the cell binding assays with H226, ^111^In-labeled NZ-16 showed significantly higher specific binding than ^111^In-labeled NZ-12 (*p* < 0.01). The maximum values were 29.6 ± 3.7% for NZ-16 at 1.0 × 10^7^ cells and 22.9 ± 3.3% for NZ-12 at 1.0 × 10^7^ (Figure 1a,b). The specific binding did not significantly differ between ^111^In- and ^225^Ac-labeled NZ-16 (Appendix A). The results of the competitive inhibition assay are shown in Figure 1c,d. The binding affinities (*K*_D_) of intact NZ-12 and DOTA-conjugated NZ-12 were estimated to be 1.7 and 5.7 nM, respectively (Figure 1c). Those of intact NZ-16 and DOTA-conjugated NZ-16 were estimated to be 1.8 and 2.7 nM, respectively (Figure 1d). These results indicate that the DOTA conjugation procedure decreased the affinity of NZ-16 for PDPN to a lesser extent than that of NZ-12.

### 3.2. Biodistribution of ^111^In-Labeled Antibodies in Nude Mice Bearing H226 Tumors

The biodistribution of ^111^In-labeled NZ-12 and NZ-16 in H226 tumor-bearing mice is shown in Table 1 and Figure 2. ^111^In-Labeled NZ-12 and NZ-16 gradually cleared from the blood and accumulated in the H226 tumors. The uptake of ^111^In-labeled NZ-16 in normal organs tended to be higher than that of ^111^In-labeled NZ-12, except in the liver (Table 1). The uptake of ^111^In-labeled NZ-16 in the liver was significantly lower than that of NZ-12 on days 1 and 2 post injection (*p* < 0.05 for day 1, *p* < 0.01 for day 2). The uptake of ^225^Ac-NZ-16 in normal organs was significantly higher than that of ^111^In-labeled NZ-16 (*p* < 0.01 in the brain, lung, spleen, and muscle, *p* < 0.05 in the intestine, Table 2). Tumor uptake of ^111^In-labeled NZ-16 tended to be higher than that of NZ-12 over the observation period (Figure 2). The maximal tumor uptake of ^111^In-labeled NZ-16 was significantly higher than that of ^111^In-labeled NZ-12 on day 4 post injection (15.1 ± 2.4% ID/g for NZ-16 and 10.0 ± 0.4% ID/g for NZ-12; *p* < 0.01, Figure 2). The tumor uptake of ^225^Ac-NZ-16 on day 4 was higher than that of ^111^In-labeled NZ-12 and NZ-16 (*p* < 0.01, Figure 2).

### 3.3. Dosimetry

The absorbed doses were estimated on the basis of the biodistribution studies when In-111 was replaced with Y-90 or Ac-225. Table 3 shows the estimated absorbed doses when the radiation-weighted factor was not considered. The absorbed doses of radiolabeled NZ-16 for tumors and organs tended to be higher than those of radiolabeled NZ-12, except for in the liver and kidney, although the difference between the two antibodies was not statistically significant (Table 3). The tumor-absorbed doses of ^225^Ac-labeled NZ-12 and NZ-16 were 60-fold greater than those of ^90^Y-labeled NZ-12 and NZ-16, respectively (Table 3).

The relative biologic effect (RBE) was determined by calculating the absorbed doses from treatments with ^90^Y- and ^225^Ac-labeled NZ-16 without considering the radiation-weighted factor (Table 4). The dose absorbed by tumors treated with 3.7 kBq of ^90^Y-labeled NZ-16 was 5.7-fold higher than that of tumors treated with 11.1 kBq of ^225^Ac-labeled NZ-16 (Table 4).

With regard to safety, the absorbed doses considering radiation weighting factors of 1 for ^90^Y and 5 for ^225^Ac are shown in Table 5. The absorbed dose to bone marrow from treatment with 3.7 MBq of ^90^Y-labeled NZ-16 was higher than that from treatment with 11.1kBq of ^225^Ac-labeled NZ-16 (Table 5). The absorbed doses to tumors and organs injected with 18.5 kBq of ^225^Ac-labeled NZ-16 were higher than those of tumors injected with 3.7 MBq of ^90^Y-labeled NZ-16 (Table 5).

### 3.4. Treatment Effects of Radiolabeled Antibodies in Nude Mice Bearing H226 Tumors

Marked antitumor effects were observed in mice treated with ^90^Y- and ^225^Ac-labeled NZ-16 (*p* < 0.01, vs. 0 MBq, Figure 3a). Treatment with 3.7 MBq of ^90^Y-NZ-16 reduced tumor volume from day 7 to day 21 post injection, and thereafter the tumor volume gradually increased (*p* < 0.01 vs. 0 MBq, Figure 3a and Appendix A). In the group injected with 11.1 kBq of ^225^Ac-NZ-16, the tumor volume increased during the first 10 days, and thereafter decreased until day 42 (*p* < 0.01 vs. 0 MBq, Figure 3a). In the group injected with 18.5 kBq of ^225^Ac-NZ-16, tumor growth was suppressed during the first 28 days, and thereafter the tumor volume decreased until the end of the observation period (*p* < 0.01 vs. 0 MBq and 3.7 MBq of ^90^Y-NZ-16, Figure 3a).

Kaplan–Meier survival curves based on a tumor volume endpoint of 300 mm^3^ are shown in Figure 3b. Injection with 3.7 MBq of ^90^Y-labeled NZ-16, 11.1 kBq ^225^Ac-labeled NZ-16, and 18.5 kBq of ^225^Ac-labeled NZ-16 significantly prolonged survival compared with the 0-MBq groups (*p* < 0.01). At the end of the observation period, all mice treated with 11.1 kBq and 18.5 kBq of ^225^Ac-NZ-16 were defined as surviving, and survival in the ^90^Y-treatment group was 60%. No statistically significant difference in survival was detected among the three groups treated with the radiolabeled antibodies (Figure 3b).

The three radiolabeled treatments induced temporary weight loss. Body weight loss never exceeded 20% compared with that at day 0, however, which is the criterion for humane euthanasia (Appendix A). No obvious damage was detected in the spleen, kidney, liver, or bone marrow in mice treated with ^90^Y- or ^225^Ac-labeled NZ-16 (Appendix A).

### 3.5. Histologic Analysis of H226 Tumors Treated with 90Y- and 225Ac-Labeled NZ-16

Tumors treated with 0 MBq of NZ-16 (intact NZ-16 only) were composed of solid nests of epithelial cells and some mitotic cells (Figure 4, upper panels). Sections of H226 tumors treated with 3.7 MBq of ^90^Y-labeled NZ-16 showed a few small necrotic foci on day 1, and expansion of the necrotic area was observed on days 3 and 7 (Figure 4, middle panels). In the tumors treated with 18.5 kBq of ^225^Ac-NZ-16 on day 1, more necrotic foci were observed compared with the tumors treated with ^90^Y-NZ-16, and extensive necrosis and lymphocyte infiltration were observed on day 3 post injection (Figure 4, lower panels). On day 7, the tumor cells decreased and partial replacement of necrotic tumor cells by fibrous tissue was observed in ^225^Ac-NZ-16-treated tumors (Figure 4, lower panels).

Treatment with 3.7 MBq of ^90^Y-labeled NZ-16 and 18.5 kBq of ^225^Ac-labeled NZ-16 significantly reduced the proliferation of (Ki-67-positive) tumor cells compared with tumors treated with 0 MBq NZ-16 (intact NZ-16) on days 1–7 post injection (*p* < 0.01, Figure 5a,b). A few apoptotic cells were observed in tumors treated with ^90^Y- and ^225^Ac-NZ-16, but no apoptosis was observed in the 0-MBq group (Appendix A).

## 4. Discussion

An anti-PDPN antibody, NZ-16, was newly developed from the parental antibody NZ-1. The constant region of the NZ-16 heavy chain differs from that in NZ-12, which was evaluated as a radioimmunotherapeutic agent in a previous study [6]. NZ-16 has a higher affinity and showed higher tumor uptake in a PDPN-expressing H226 mesothelioma mouse model than NZ-12. Therefore, RIT with NZ-16 was expected to be more effective and NZ-16 was selected for further evaluation. As expected, ^90^Y- and ^225^Ac-labeled NZ-16 showed significant antitumor effects in tumor-bearing mice, compared with ^90^Y-labeled NZ-12 [6]. The significantly higher effectiveness of ^225^Ac-labeled NZ-16 compared with ^90^Y-labeled NZ-16 (*p* < 0.01) suggests that α-RIT with NZ-16 is a promising therapy for malignant mesothelioma. Our findings are encouraging and warrant further studies toward clinical applications.

PDPN is highly expressed in many types of cancer, such as brain tumors [11], squamous cell carcinoma [25], soft tissue tumors [26], and bladder cancer [27]. A preliminary study showed radiolabeled NZ-16 highly bound to PDPN-expressing LN-319 glioma cells and showed high tumor uptake. RIT with ^225^Ac-labeled NZ-16 is applicable for the treatment of such cancers, although further preclinical studies in these cancer models are required.

Our pathologic analysis showed that ^225^Ac-labeled NZ-16 induced a larger extent of necrotic change in tumor tissues compared with ^90^Y-labeled NZ-16, although the extent of apoptotic cell death and reduction in proliferating cells were similar. α-Emitters can provide a large amount of energy and induce irreparable damage to cells, resulting in more cell death, manifested as apoptosis or necrosis, compared with β-emitters [13]. Our findings revealed that ^225^Ac-labeled NZ-16 more frequently induced necrosis than apoptosis. This result is consistent with previous reports of RIT with α-emitters that apoptotic change is not often observed in solid tumors [28]. Further studies with various cancer types are needed to evaluate whether α-particle-induced cell death is depending on the cancer type.

In the present study, the estimated tumor absorbed doses following treatment with 3.7 MBq of ^90^Y-labeled NZ-16 and 11.1 kBq of ^225^Ac-labeled NZ-16 were 14.9 and 2.6 Gy, respectively (Table 4). Our finding indicates that the RBE of ^225^Ac-labeled NZ-16 is 5.7, which is similar to the recommended amount for α-emitters by the Medical Internal Radiation Dose Committee [24]. The treatment with ^225^Ac-labeled NZ-16 was markedly more effective than that with ^90^Y-labeled NZ-16, suggesting that the real RBE is greater than 5.7. To estimate the RBE by another method, we referred to the results of X-ray treatment against H226 tumors in a previous report [6]. The efficacy of 11.1 kBq of ^225^Ac-labeled NZ-16 was almost equivalent to that of 50 Gy of X-ray radiation, and the RBE was calculated to be 19.2 for ^225^Ac relative to X-rays. The efficacy of 3.7 MBq of ^90^Y-labeled NZ-16 was equivalent to 25 Gy of X-ray irradiation, and the RBE was calculated to be 1.7 for ^90^Y relative to X-rays. Taken together, the RBE for ^225^Ac to ^90^Y was calculated to be 11.3. The RBE values for the cell-killing effects of α-emitters are reported to be three to five on the basis of in vivo experiments [24]. Our previous study with α-emitting ^211^At-MABG also showed that the RBE was approximately three [28]. Compared with other α-emitting compounds, the calculated RBE for ^225^Ac-labeled NZ-16 is quite high. The reason for this discrepancy is not clear at present, but the increased RBE might depend on the tumor type. Further studies are necessary to estimate a more accurate RBE for ^225^Ac-labeled NZ-16 to predict the therapeutic efficacy and safety in patients.

The clinical safety of RIT with NZ-16 must be evaluated. Bone marrow is generally the dose-limiting tissue in RIT. The limiting absorbed doses in the bone marrow are 6–9 Sv in rodents and 4.5 Sv in humans [23,29]. We provided the estimated absorbed doses of ^90^Y- and ^225^Ac-labeled NZ-16, considering a radiation weighting factor of one for ^90^Y and five for ^225^Ac [24] in Table 5. The dose to the bone marrow was 8.1 Sv_RBE1_ for 3.7 MBq of ^90^Y-labeled NZ-16 and 7.0 Sv_RBE5_ for 11.1 kBq of ^225^Ac-labeled NZ-16. These doses would be acceptable in rodents; treatment-related mortality and toxicity to the bone marrow were not observed in the present study, although there was a temporary decrease in body weight. The doses, however, are greater than the limiting dose in humans of 4.5 Sv. The injected dose to patients should thus be decreased. The biodistribution of drugs, including antibodies, however, is generally not identical between humans and animals. Clinical dosimetry studies are needed to determine a safe injected radioactive dose for humans.

In mesothelioma patients, most mesothelioma cells spread into the diaphragm, chest wall, and mediastinum; radiotherapy is therefore limited due to the high risk of injury to the lungs and surrounding organs [30]. Radiation pneumonitis is the most common toxicity in patients treated with radiation for cancers in the thorax [31]. The mean dose to the lungs for a 20% risk of radiation pneumonitis is 20 Sv [31]. Our dosimetry showed that the absorbed doses of ^90^Y- and ^225^Ac-labeled NZ-16 in the lungs were lower than 20 Sv. The spleen, liver, and kidneys would also be tolerable because these doses were lower than the tolerated doses in humans [31]. Therefore, the risk of radiation-induced toxicity from RIT with radiolabeled NZ-16 is expected to be low.

As a SPECT imaging agent for dosimetry and treatment monitoring, ^111^In-labeled antibodies are a suitable surrogate for ^225^Ac-labeled antibodies [32]. In the present study, uptake of ^225^Ac-labeled NZ-16 in most organs tended to be higher than that of ^111^In-labeled NZ-16, and the RBE of ^225^Ac-labeled NZ-16 on tumor growth suppression was greater than five. Taken together, the dosimetry of ^225^Ac-labeled NZ-16 could be underestimated by ^111^In-labeled surrogate imaging. Although the present study revealed no severe damage in mice treated with ^225^Ac-labeled NZ-16, further studies, including dosimetric assessments of toxicity in the therapeutic use of α-RIT for solid tumors, are needed to determine the appropriate dose of ^225^Ac-labeled NZ-16 for first-in-human studies with high confidence.

The present study has several limitations. First, ^225^Ac-labeled NZ-16 did not achieve complete remission. Further strategies are needed to improve the antitumor effect. Fractionated therapy with ^225^Ac-labeled NZ-16 is promising. Two fractions of ^225^Ac-lintuzumab produced complete remission in patients with hematologic malignancies [33]. The therapeutic effects of RIT might be enhanced by combining them with chemotherapeutic agents. Pemetrexed, as a first-line chemotherapy for mesothelioma, has a radiosensitizing effect and might therefore be particularly effective with ^225^Ac-labeled NZ-16 [34]. Second, our dosimetry in mice cannot be directly applied to determine the appropriate dose for malignant mesothelioma patients. Clinical imaging studies with ^111^In-labeled NZ-16 are required to guarantee the safety of therapeutic treatments. These studies would promote the possible clinical application of ^225^Ac-labeled NZ-16 for malignant mesothelioma. Third, the present study employed only one PDPN-expressing mesothelioma cell line. Unfortunately, another mesothelioma cell line with high PDPN expression is not available. There is a need to genetically develop PDPN-expressing mesothelioma cell lines and evaluate the efficacy of radiolabeled NZ-16 in the future.

## 5. Conclusions

The novel anti-PDPN antibody NZ-16 has a higher binding affinity and higher tumor uptake compared with the previous antibody NZ-12. Treatment with ^225^Ac-labeled NZ-16 showed a potent antitumor effect without obvious adverse effects in a mesothelioma mouse model. RIT with ^225^Ac-labeled NZ-16 is a promising potential therapeutic option for malignant mesothelioma patients.

## Figures and Tables

**Figure 1 cells-10-02503-f001:**
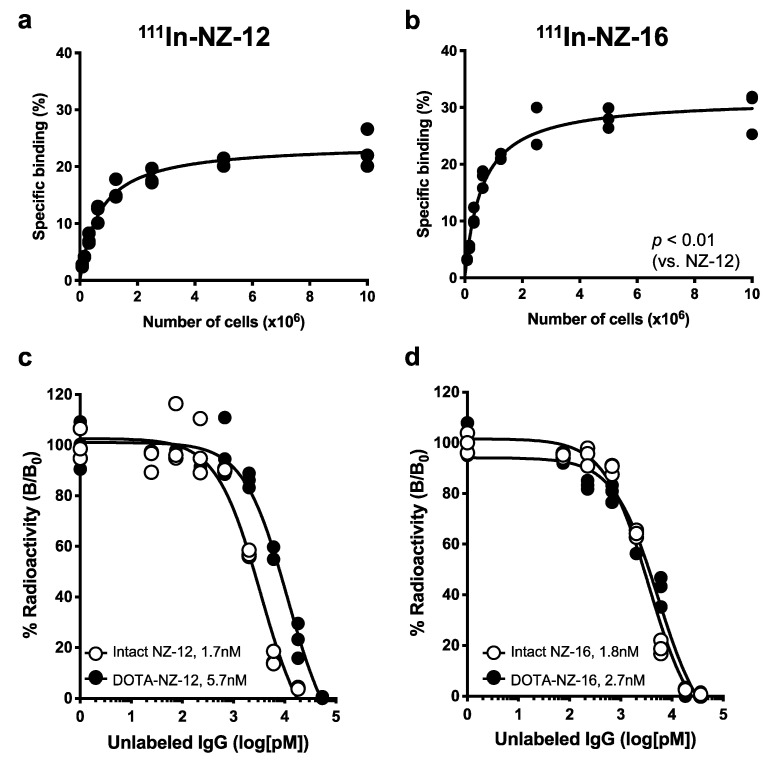
In vitro characterization of radiolabeled antibodies NZ-12 and NZ-16. (**a**) Cell binding assay of ^111^In-labeled NZ-12 with H226 cells. (**b**) Cell binding assay of ^111^In-labeled NZ-16 with H226 cells. (**c**) Competitive inhibition assay for intact NZ-12 (white circles) and DOTA-conjugated NZ-12 (black circles) with H226 cells. (**d**) Competitive inhibition assay for intact NZ-16 (white circles) and DOTA-conjugated NZ-16 (black circles) with H226 cells.

**Figure 2 cells-10-02503-f002:**
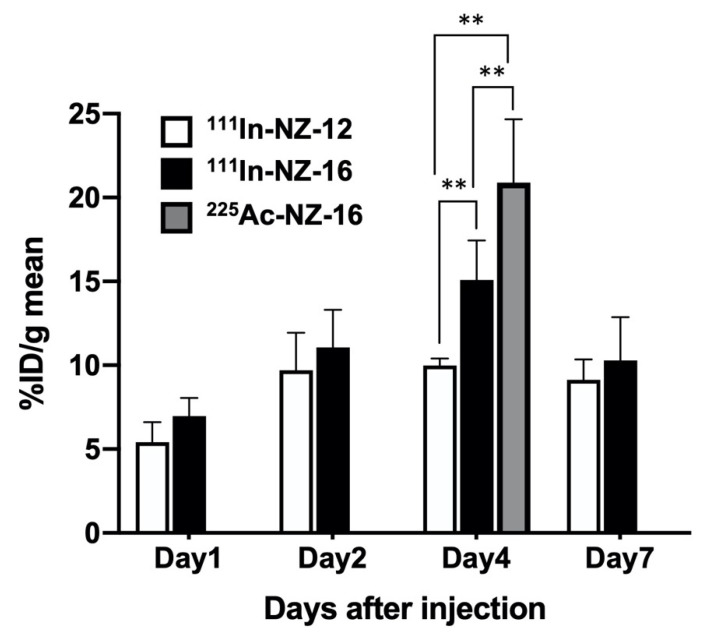
Tumor uptake of ^111^In-labeled NZ-12 (N = 4), and ^111^In- labeled NZ-16 (N = 5) and ^225^Ac-labeled NZ-16 (N = 5). Data indicate the mean and standard deviation (N = 4–5). ** *p* < 0.01.

**Figure 3 cells-10-02503-f003:**
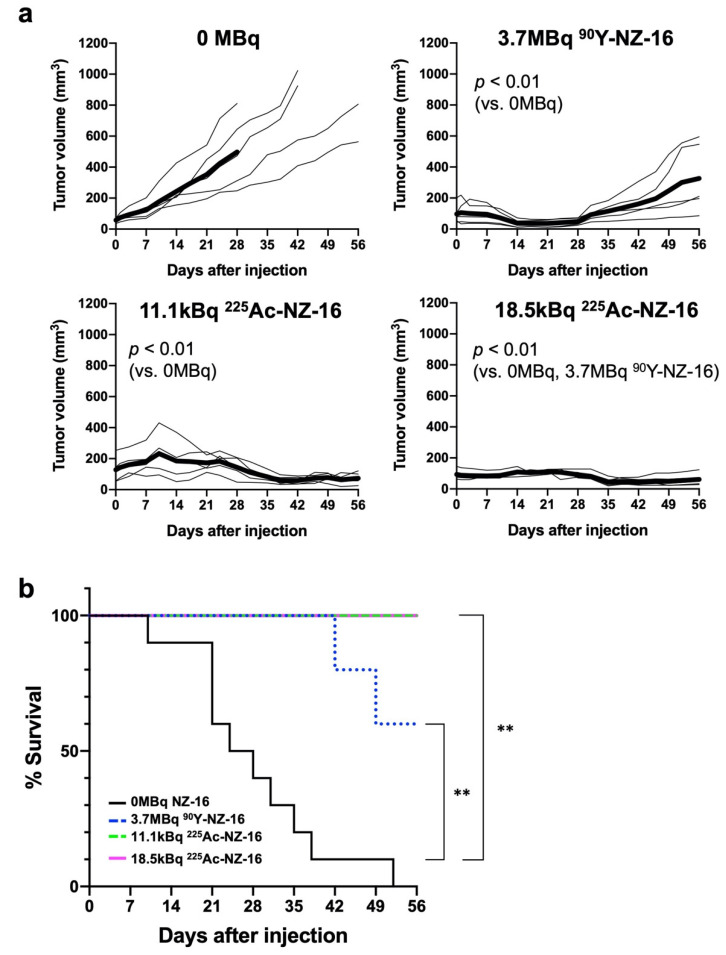
Therapeutic efficacy of ^90^Y- and ^225^Ac-labeled antibodies in H226 tumor-bearing mice (N = 5/treatment group). (**a**) Tumor growth curves of individual mice after injection with ^90^Y- or ^225^Ac-labeled NZ-16 (thin lines). Bold lines indicate the mean. (**b**) Kaplan–Meier survival curves based on a tumor volume endpoint of 300 mm^3^. ** *p* < 0.01, vs. 0 MBq NZ-16.

**Figure 4 cells-10-02503-f004:**
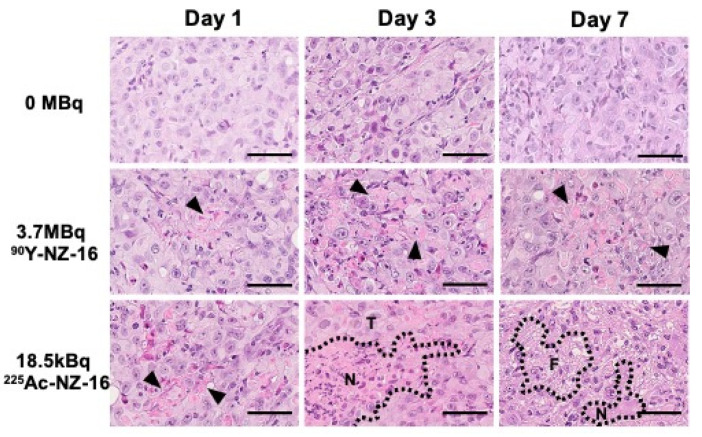
Hematoxylin and eosin-stained sections of H226 tumors treated with 0 MBq (intact NZ-16 only), 3.7 MBq of ^90^Y-labeled NZ-16, and 18.5 kBq of ^225^Ac-labeled NZ-16 at days 1, 3, and 7 after injection. Arrowheads indicate necrosis. N, necrosis; T, tumor; F, fibrous tissue; bars, 50 μm.

**Figure 5 cells-10-02503-f005:**
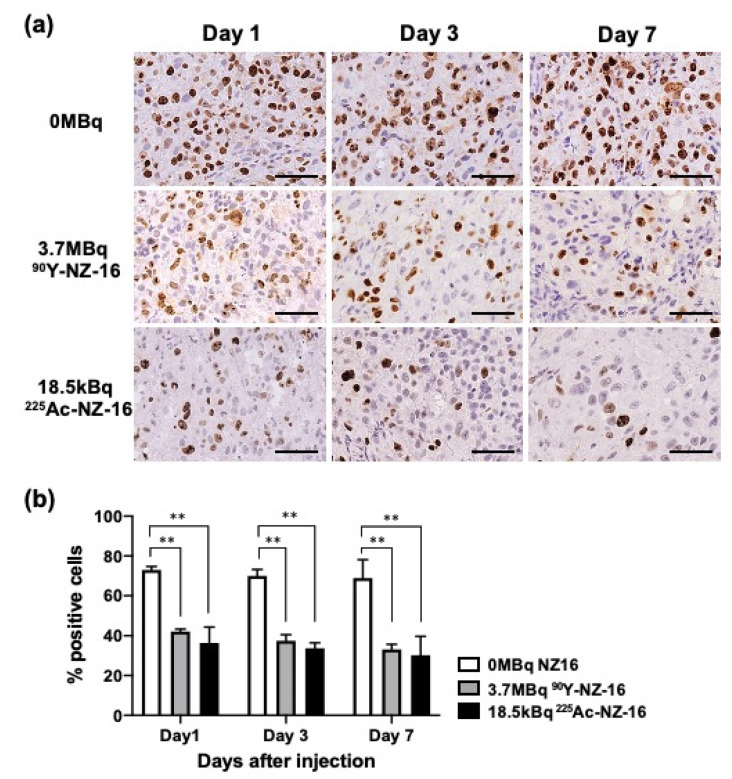
Tumor cell proliferation analysis using Ki-67 immunostaining. (**a**) Ki-67-stained H226 tumors at days 1, 3, and 7 after injection with 0 MBq (intact NZ-16 only), 3.7 MBq of ^90^Y-labeled NZ-16, and 18.5 MBq ^225^Ac-labeled NZ-16. Bar, 50 μm (**b**) Quantification of proliferating (Ki-67 positive) cells (N = 3/group). Data represent the mean and standard deviation. ** *p* < 0.01. is a figure. Schemes follow the same formatting.

**Table 1 cells-10-02503-t001:** Biodistribution of ^111^In-labeled antibodies in H226 tumor-bearing mice.

	Day 1	Day 2	Day 4	Day 7
NZ-12				
Blood	18.0 ± 1.5	13.8 ± 1.5	8.6 ± 2.3	5.1 ± 0.5
Brain	0.4 ± 0.1	0.4 ± 0.0	0.3 ± 0.1	0.2 ± 0.1
Lung	6.2 ± 0.4	5.0 ± 1.0	3.6 ± 1.0	2.1 ± 0.3
Liver	15.2 ± 2.1	14.1 ± 1.1	10.8 ± 1.0	7.3 ± 0.9
Spleen	9.9 ± 0.7	9.0 ± 1.4	8.6 ± 0.9	5.4 ± 1.4
Intestine	2.0 ± 0.3	1.6 ± 0.2	1.0 ± 0.2	0.6 ± 0.1
Kidney	12.1 ± 0.3	9.5 ± 0.6	6.4 ± 1.0	3.7 ± 0.7
Muscle	0.9 ± 0.1	0.9 ± 0.1	0.9 ± 0.2	0.3 ± 0.1
NZ-16				
Blood	21.8 ± 2.9	19.7 ± 2.6 **	13.6 ± 1.4 *	8.2 ± 4.0
Brain	0.6 ± 0.1	0.6 ± 0.1	0.5 ± 0.1	0.3 ± 0.2
Lung	7.7 ± 1.3	6.9 ± 0.7 *	7.0 ± 2.7	3.4 ± 1.4
Liver	11.3 ± 3.5 *	9.3 ± 0.9 **	7.9 ± 2.3	7.1 ± 1.6
Spleen	7.5 ± 0.8	9.7 ± 0.7	8.0 ± 0.9	8.1 ± 3.1 *
Intestine	2.6 ± 0.6	2.2 ± 0.2	1.9 ± 0.4	1.0 ± 0.4
Kidney	7.9 ± 1.2 **	7.8 ± 0.7	7.0 ± 0.7	4.9 ± 1.6
Muscle	1.2 ± 0.2	1.2 ± 0.3 *	1.1 ± 0.1	0.7 ± 0.4

Data are indicated as the percentage of injected dose per gram (% ID/g) and as the mean ± standard deviation. * *p* < 0.05, ** *p* < 0.01 vs. NZ-12.

**Table 2 cells-10-02503-t002:** Biodistribution of ^225^Ac-labeled NZ-16 in H226 tumor-bearing mice.

	Day 4
Blood	12.4 ± 0.8
Brain	2.0 ± 0.4 **
Lung	9.9 ± 1.1 **
Liver	10.5 ± 1.5
Spleen	11.9 ± 1.0 **
Intestine	2.7 ± 0.6 *
Kidney	7.9 ± 1.0
Muscle	3.6 ± 1.1 **

Data are indicated as the percentage of injected dose per gram (% ID/g) and as the mean ± standard deviation. ** *p* < 0.01, * *p* < 0.05 vs. ^111^In-NZ-16 (Table 1).

**Table 3 cells-10-02503-t003:** Estimated absorbed dose (Gy/MBq) for ^90^Y- and ^225^Ac-labeled NZ-12 and NZ-16 based on the biodistribution data of ^111^In-labeled antibodies, not considering a radiation weighting factor.

	^90^Y		^225^Ac	
	NZ-12	NZ-16	NZ-12	NZ-16
Brain	0.1 ± 0.0	0.2 ± 0.0	6.4 ± 0.5	9.7 ± 0.8
Lung	1.7 ± 0.1	2.4 ± 0.2	84.6 ± 4.7	124.2 ± 11.4
Liver	4.6 ± 0.1	3.5 ± 0.2	239.0 ± 7.3	185.5 ± 13.5
Spleen	3.2 ± 0.1	3.1 ± 0.1	168.7 ± 9.4	172.1 ± 36.4
Intestine	0.5 ± 0.0	0.7 ± 0.0	25.8 ± 1.1	37.1 ± 2.0
Kidney	3.1 ± 0.1	2.6 ± 0.1	156.9 ± 4.7	140.0 ± 9.5
Muscle	0.3 ± 0.0	0.4 ± 0.0	15.5 ± 0.9	21.1 ± 1.7
Bone marrow ^a^	1.8 ± 0.1	2.5 ± 0.1	89.4 ± 4.0	126.1 ± 6.1
Tumor	3.0 ± 0.2	3.8 ± 0.2	174.6 ± 158.0	236.0 ± 108.6

Data indicate the mean ± standard deviation. ^a^ The absorbed doses of bone marrow were estimated based on the blood uptake (Table 1), considering a red-marrow-to-blood activity ratio of 0.4. There is no significant difference between NZ-12 and NZ-16.

**Table 4 cells-10-02503-t004:** Estimated absorbed doses (Gy) from the treatment dose of ^90^Y- and ^225^Ac-labeled NZ-16, not considering a radiation weighting factor.

	3.7 MBq ^90^Y	11.1 kBq ^225^Ac	18.5 kBq ^225^Ac
Brain	0.7	0.1	0.2
Lung	8.8	1.4	2.3
Liver	12.8	2.1	3.4
Spleen	11.4	1.9	3.2
Intestine	2.7	0.4	0.7
Kidney	9.7	1.6	2.6
Muscle	1.5	0.2	0.4
Bone marrow ^a^	8.1	1.4	2.3
Tumor	14.9	2.6	4.4

^a^ The absorbed doses of bone marrow were estimated based on the blood uptake (Table 1), considering a red-marrow-to-blood activity ratio of 0.4.

**Table 5 cells-10-02503-t005:** Estimated absorbed doses (Sv) from the treatment dose of ^90^Y- and ^225^Ac-labeled NZ-16 using radiation weighting factors ^a^.

	3.7 MBq ^90^Y	11.1 kBq ^225^Ac	18.5 kBq ^225^Ac
Brain	0.7	0.5	0.9
Lung	8.8	6.9	11.5
Liver	12.8	10.3	17.2
Spleen	11.4	9.6	15.9
Intestine	2.7	2.1	3.4
Kidney	9.7	7.8	13.0
Muscle	1.5	1.2	2.0
Bone marrow ^b^	8.1	7.0	11.7
Tumor	14.9	13.1	21.8

^a^ The radiation weighting factors of 1 for ^90^Y and 5 for ^225^Ac. ^b^ The absorbed doses of bone marrow were estimated based on the blood uptake (Table 1), considering a red-marrow-to-blood activity ratio of 0.4.

## Data Availability

The data presented in this study are available on request from the corresponding author.

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
