# Peer review of "Preclinical Evaluation of Podoplanin-Targeted Alpha-Radioimmunotherapy with the Novel Antibody NZ-16 for Malignant Mesothelioma"

_cells, 2021, doi:10.3390/cells10102503_

Round 1

Reviewer 1 Report

The present manuscript shows the preclinical evaluation of a new antibody against mesothelioma. Concretely, an antibody against Podoplanin glycoprotein, a protein overexpressed in mesothelioma. The authors refer to a previous published paper of their own (Sudo, H et al. Therapeutic efficacy evaluation of radioimmunotherapy with 90Y-labeled anti-podoplanin antibody NZ-12 for mesothelioma. Cancer Sci 2019, 110, 1653-1664, doi:10.1111/cas.13979), in which they demonstrated that 90Y-labeled anti-podoplanin antibody NZ-12 was efficient against mesothelioma. The present study shows that changing NZ-12 for NZ-16, the efficacy against mesothelioma is higher. So, the authors conclude that NZ-16 is the antibody to be chosen for these treatments.

MAJOY ISSUES

  1. The major issue against acceptance of this manuscript would be that only one mesothelioma cell line has been used in the study, while, in the previous paper on NZ-12 the same authors included –at least- two cell lines.
  2. No images of mice bearing tumors are included.
  3. Number of mice has not been included.
  4. Please, define MBq. The reviewer did not find it in the manuscript.
  5. Please, define separately, more clearly In-labeled, Y-labeled, Ac-labeled, DOTA-conjugated, for the common reader of cancer biology not so familiarized with antibody labeling.
  6. Figure 2. Please, explain why Ac-NZ-16 only appears, suddenly, in day 4. And why In-NZ-12 and 16 are always present.
  7. The discussion (390 to 421) is rather lengthy and not so clear for usual readers of cancer biology.
  8. Immunofluorescence of cells showing positivity for NZ-16 is not included.
  9. Mesothelioma human specimens are not included to show expression of podoplanin.
  10. Please, include mesothelioma most typical genetic markers in the Introduction.

Author Response

Dear Reviewer 1,

We greatly appreciate the reviewer's helpful comments and suggestions, which we found very useful in revising our manuscript and in planning further studies. Our point-by-point responses to the reviewer's comments are shown below. We used the track change mode and the highlight function in MS Word to indicate all changes in our revised manuscript.

  1. The major issue against acceptance of this manuscript would be that only one mesothelioma cell line has been used in the study, while, in the previous paper on NZ-12 the same authors included –at least- two cell lines.

Reply: We agree that is a major point. As we reported, our anti-PDPN antibodies derived from NZ-1 do not bind to any PDPN-negative cell lines (references below). In line with the findings, no PDPN-negative cell line is necessary for the present study. There is no PDPN-positive cell line that develops xenograft tumors, except for H226, the present study therefore could employ only the one cell line H226. However, the present study was to show the superiority of 225Ac-NZ-16, which is provided in the current submission. We believe that the reviewer will agree with that.

As the reviewer suggests, there is a need to evaluate the efficacy in different PDPN-positive tumor models. Actually, we have found another cell line expressing PDPN derived from glioma. Radiolabeled NZ-16 binds highly to the cell line. We expect 225Ac-labeled NZ-16 is highly effective to the glioma cell line as well as H226. Unfortunately, the cell line is not derived from mesothelioma, therefore, we cannot include it in the present study. Now, we are planning to evaluate the efficacy of 225Ac-labeled NZ-16 in the glioma model as another study.

This limitation was added to the Discussion section as a limitation as follows: "Third, the present study employed only one PDPN-expressing mesothelioma cell line. Unfortunately, another mesothelioma cell line with high PDPN expression is not available. There is a need to genetically develop PDPN-expressing mesothelioma cell lines and evaluate the efficacy of radiolabeled NZ-16 in the future."

Refs: Sudo, H.; Tsuji, A.B.; Sugyo, A.; Saga, T.; Kaneko, M.K.; Kato, Y.; Higashi, T. Therapeutic efficacy evaluation of radioimmunotherapy with 90Y-labeled anti-podoplanin antibody NZ-12 for mesothelioma. Cancer Sci 2019, 110, 1653-1664, doi:10.1111/cas.13979.

Abe, S.; Morita, Y.; Kaneko, M.K.; Hanibuchi, M.; Tsujimoto, Y.; Goto, H.; Kakiuchi, S.; Aono, Y.; Huang, J.; Sato, S., et al. A novel targeting therapy of malignant mesothelioma using anti-podoplanin antibody. J Immunol 2013, 190, 6239-6249, doi:10.4049/jimmunol.1300448.

  1. No images of mice bearing tumors are included.

Reply: We provided mouse images as Supplementary Figure 3 and added the following sentences "Marked antitumor effects were observed in mice treated with 90Y- and 225Ac-labeled NZ-16 (P < 0.01, vs. 0 MBq, Figures 3a and S3)." to Section 3.4.

  1. Number of mice has not been included.

Reply:  We added the numbers of mice as follows: “When tumor volumes reached approximately 50 mm3, mice (n = 4–5/time-point), were intravenously injected with 111In-labeled NZ-12 (37 kBq, n = 4/time-point), 111In-labeled NZ-16 (37 kBq, n = 5/time-point), or 225Ac-labeled NZ-16 (3.7 kBq, n = 5/time-point) in a total of 40 mg antibody adjusted by adding the corresponding unlabeled antibodies.” in Section 5.6, “Tumor uptake of 111In-labeled NZ-12 (n = 4), and 111In-labeled NZ-16 (n = 5) and 225Ac-labeled NZ-16 (n = 5).” in the Figure 2 legend, “Therapeutic efficacy of 90Y- and 225Ac-labeled antibodies in H226 tumor-bearing mice (n = 5/treatment group).” in the Figure 3 legend, and “Quantification of proliferating (Ki-67 positive) cells (n = 3/group).” in the Figure 5 legend.

  1. Please, define MBq. The reviewer did not find it in the manuscript.

Reply: MBq means megabecquerel, which was added to Section 2.8.

  1. Please, define separately, more clearly In-labeled, Y-labeled, Ac-labeled, DOTA-conjugated, for the common reader of cancer biology not so familiarized with antibody labeling.

Reply: To avoid common readers' inconvenience, we added the following sentences "For radiolabeling of radiometals 111In, 90Y, and 225Ac, antibodies are necessary to be conjugated with a chelator. The present study employed p-SCN-Bn-DOTA (DOTA, Macrocyclics, Dallas, TX, USA). All radiolabeled antibodies were conjugated with DOTA, but DOTA was abbreviated; namely, we presented them like 111In-, 90Y-, and 225Ac-labeled antibodies." to Section 2.3.

  1. Figure 2. Please, explain why Ac-NZ-16 only appears, suddenly, in day 4. And why In-NZ-12 and 16 are always present.

Reply: It is difficult for us to obtain 225Ac, which forced us to plan minimal experiments with 225Ac. We therefore conducted the biodistribution study of 225Ac-labeled NZ-16 on only day 4, which was the day observed the highest tumor uptake of 111In-labeled NZ-16. The reason why 111In-labeled NZ-12 and NZ-16 are always present is to provide the head-to-head comparison.

  1. The discussion (390 to 421) is rather lengthy and not so clear for usual readers of cancer biology.

Reply: In accordance with the reviewer’s comment, we revised them as follows: "In mesothelioma patients, most mesothelioma cells spread into the diaphragm, chest wall, and mediastinum; radiotherapy is therefore limited due to the high risk of injury to the lungs and surrounding organs [30]. Radiation pneumonitis is the most common toxicity in patients treated with radiation for cancers in the thorax [31]. The mean dose to the lungs for a 20% risk of radiation pneumonitis is 20 Sv [31]. Our dosimetry showed that the absorbed doses of 90Y- and 225Ac-labeled NZ-16 by the lungs were lower than 20 Sv. The spleen, liver, and kidneys would also be tolerable because these doses were lower than the tolerated doses in humans [31]. Therefore, the risk of radiation-induced toxicity from RIT with radiolabeled NZ-16 is expected to be low." and "As a SPECT imaging agent for dosimetry and treatment monitoring, 111In-labeled antibodies are a suitable surrogate for 225Ac-labeled antibodies [32]. In the present study, uptake of 225Ac-labeled NZ-16 in most organs tended to be higher than that of 111In-labeled NZ-16, and the RBE of 225Ac-labeled NZ-16 on tumor growth suppression was greater than 5. Taken together, the dosimetry of 225Ac-labeled NZ-16 could be underestimated by 111In-labeled surrogate imaging. Although the present study revealed no severe damage in mice treated with 225Ac-labeled NZ-16, further clinical studies are needed to determine the appropriate dose of 225Ac-labeled NZ-16 for first-in-human studies with high confidence."

  1. Immunofluorescence of cells showing positivity for NZ-16 is not included.

Reply: We provided the immunofluorescence staining images as new Supplementary Figure 1, and we added the following sentences "To confirm the reactivity of NZ-16 in H226 cells, immunofluorescence staining was conducted. A strong intensity was observed on the cell membranes of H226 (Figure S1)." to Section 3.1.

  1. Mesothelioma human specimens are not included to show expression of podoplanin.

Reply: We have reported that the expression analysis of PDPN in human specimens (references below). The present study therefore includes no data but includes that information in the Introduction section of the revised manuscript.

Refs: Nishinaga, Y.; Sato, K.; Yasui, H.; Taki, S.; Takahashi, K.; Shimizu, M.; Endo, R.; Koike, C.; Kuramoto, N.; Nakamura, S., et al. Targeted Phototherapy for Malignant Pleural Mesothelioma: Near-Infrared Photoimmunotherapy Targeting Podoplanin. Cells 2020, 9, doi:10.3390/cells9041019.

Sudo, H.; Tsuji, A.B.; Sugyo, A.; Saga, T.; Kaneko, M.K.; Kato, Y.; Higashi, T. Therapeutic efficacy evaluation of radioimmunotherapy with 90Y-labeled anti-podoplanin antibody NZ-12 for mesothelioma. Cancer Sci 2019, 110, 1653-1664, doi:10.1111/cas.13979.

  1. Please, include mesothelioma most typical genetic markers in the Introduction.

Reply: As per the reviewer's comment, we added the following sentences "Mesothelioma is classified into three types, epithelioid, sarcomatoid, and biphasic, based on histological characteristics [1,2]. There are several markers for the epithelioid subtype, such as calretinin, WT-1, cytokeratin 5, and ERC/mesothelin [3,4]. Those markers do not express in the sarcomatoid subtype, but podoplanin (PDPN) is overexpressed in more than 80% of all the types [5,6]."

Reviewer 2 Report

Re: “Preclinical evaluation of podoplanin-targeted alpha-radio-immunotherapy with the novel antibody NZ-16 for malignant 3 mesothelioma”

This submitted manuscript introduces a “chimeric” version of a previous studied rat monoclonal antibody targeting human podoplanin (PDPN), a molecule overexpressed in malignant mesothelioma (in not all cases) but also expressed by normal tissues. Even though the authors tried to demonstrate an improved therapeutically antibody, it is practically impossible to assess the benefit of the chimeric vs. rat antibody based on the data presented in manuscript. There are also so many controls missing in the experimental part of the manuscript. Therefore, the manuscript needs a major re-designing of the whole methodology and a much better interpretation/integration of the data in the context of actual and real world of immunotherapy with whole monoclonal antibodies and, very important, the place of radio-immunotherapy using alpha emitters for treatment of solid tumors.

Major comments:

  1. The design of the “novel chimera” antibody and the conclusions of increasing affinity based on their antibody engineering methodology is indeed puzzling. There is no scientific discussion/explanation of how a chimeric antibody which replaces ONLY the heavy chain constant domain (aka, CH1-CH3) will impact the binding affinity to the PARATOPE in the VH-VL region of the antibody which is comprised of variables regions of heavy and light chain. Claiming that the “DOTA conjugation procedure decreased the affinity of NZ-16 for PDPN to a lesser extent than that of NZ-12” without doing any changes (including VH-VL or scFV affinity experiments) in the paratope does not have any scientific support.
  2. Replacing the rat Fc with human Fc has be followed by experiments showing the successfully “transplant” of human Fc.
  3. While preliminary data are acceptable in nude mice, the authors should have been done in vivo experiments (including bio distribution and therapy) using syngeneic models. Both rat and human constant fragment of the antibody have a huge affinity to Fc receptors positive cell lines (B cells, granulocytes, dendritic cells, natural killer cells). The main reason of the drawback for radio-immunoconjugates with alpha-emitters (and not only for them) as payloads have been the specificity of whole IgG to Fc receptors. This is the reason the last decades research has focuses on antibodies with engineering Fc receptors with short half-live or, even better, with Fc region.
  4. Using only ONE cell line for this type of study is inappropriate. At least two cell PDPN-positive lines should have been included with also a PDPN negative tumor (preferable mesothelioma origin).
  5. Did the authors check if their used tumor cells line expresses ectopically Fc receptors, even at low level, but enough to show difference in “affinity” (as demonstrated in many human tumors) which would have been make the difference since mouse/rat does not bind to human Fc?
  6. While preliminary data using subcutaneously (s.c.) tumor may be accepted, due to the unique natural history of malignant mesothelioma, the s.c. model is not appropriate to demonstrate toxicity and, much more important, therapeutically efficacy.
  7. No information regarding the immunogenicity of the chimera antibody are provide (due to the foreign parts, “chimera” part and the paratope of the chimeric antibody).
  8. No studies are shown with isotype antibody labeled to the alpha-emitters and 225Ac-lebeled NZ-12 antibody.
  9. No information regarding mycoplasma assessing and authentication of cells are provided.

Author Response

Dear Reviewer 2,

We greatly appreciate the reviewer's helpful comments and suggestions, which we found very useful in revising our manuscript and in planning further studies. Our point-by-point responses to the reviewer's comments are shown below. We used the track change mode and the highlight function in MS Word to indicate all changes in our revised manuscript.

Point-by-point responses to the comments of Reviewer 2

  1. The design of the “novel chimera” antibody and the conclusions of increasing affinity based on their antibody engineering methodology is indeed puzzling. There is no scientific discussion/explanation of how a chimeric antibody which replaces ONLY the heavy chain constant domain (aka, CH1-CH3) will impact the binding affinity to the PARATOPE in the VH-VL region of the antibody which is comprised of variables regions of heavy and light chain. Claiming that the “DOTA conjugation procedure decreased the affinity of NZ-16 for PDPN to a lesser extent than that of NZ-12” without doing any changes (including VH-VL or scFV affinity experiments) in the paratope does not have any scientific support.

Reply: We also don't know the reason why the conjugation provided less damage to NZ-16 compared with NZ-12. From our experiences, the other NZ subtypes having the same scFv and different constant regions were received much damage compared with NZ-12. Surprisingly, some subtypes showed no binding to PDPN-expressing cells after DOTA conjugation, although they bound it before the conjugation. We cannot provide direct evidence but there is a possible reason: some DOTA conjugated at some positions of the constant domain might inhibit the binding of some NZ subtypes to PDPN. The DOTA derivative used in the present study conjugates to lysine. Damage induced by conjugation might depend on the positions of lysine in the constant domain. Although we don't know the reason, we know constant regions affect the binding of our NZ antibodies. We strongly want to select the best one to apply to clinical trials. Exploring more than thirty subtypes found NZ-16 the best in vitro. The present study was conducted to obtain preclinical evidence in vivo. Our findings encourage further clinical studies.

  1. Replacing the rat Fc with human Fc has be followed by experiments showing the successfully “transplant” of human Fc.

Reply: As the reviewer mentioned, we confirmed the replacement was successful.

  1. While preliminary data are acceptable in nude mice, the authors should have been done in vivo experiments (including bio distribution and therapy) using syngeneic models. Both rat and human constant fragment of the antibody have a huge affinity to Fc receptors positive cell lines (B cells, granulocytes, dendritic cells, natural killer cells). The main reason of the drawback for radio-immunoconjugates with alpha-emitters (and not only for them) as payloads have been the specificity of whole IgG to Fc receptors. This is the reason the last decades research has focuses on antibodies with engineering Fc receptors with short half-live or, even better, with Fc region.

Reply: We agree with the reviewer. The amount of protein using RIT is very small compared to general immunotherapy, suggesting that the effect on immune cells is small. However, as the reviewer suggested, it is necessary to confirm the effect on Fc receptor-positive cells, and we will plan to conduct further preclinical studies using syngeneic animal models in the future.

  1. Using only ONE cell line for this type of study is inappropriate. At least two cell PDPN-positive lines should have been included with also a PDPN negative tumor (preferable mesothelioma origin).

Reply: We agree that is a major point. As we reported, our anti-PDPN antibodies derived from NZ-1 do not bind to any PDPN-negative cell lines (references below). In line with the findings, no PDPN-negative cell line is not necessary for the present study. There is no PDPN-positive cell line that develops xenograft tumors, except for H226, the present study therefore could employ only the one cell line H226. However, the present study was to show the superiority of 225Ac-NZ-16, which is provided in the current submission. We believe that the reviewer will agree with that.

As the reviewer suggests, there is a need to evaluate the efficacy in different PDPN-positive tumor models. Actually, we have found another cell line expressing PDPN derived from glioma. Radiolabeled NZ-16 binds highly to the cell line. We expect 225Ac-labeled NZ-16 is highly effective to the glioma cell line as well as H226. Unfortunately, the cell line is not derived from mesothelioma, therefore, we cannot include it in the present study. Now, we are planning to evaluate the efficacy of 225Ac-labeled NZ-16 in the glioma model as another study.

This was added to the Discussion section as a limitation as follows: "Third, the present study employed only one PDPN-expressing mesothelioma cell line. Unfortunately, another mesothelioma cell line with high PDPN expression is not available. There is a need to genetically develop PDPN-expressing mesothelioma cell lines and evaluate the efficacy of radiolabeled NZ-16 in the future."

Refs: Sudo, H.; Tsuji, A.B.; Sugyo, A.; Saga, T.; Kaneko, M.K.; Kato, Y.; Higashi, T. Therapeutic efficacy evaluation of radioimmunotherapy with 90Y-labeled anti-podoplanin antibody NZ-12 for mesothelioma. Cancer Sci 2019, 110, 1653-1664, doi:10.1111/cas.13979.

Abe, S.; Morita, Y.; Kaneko, M.K.; Hanibuchi, M.; Tsujimoto, Y.; Goto, H.; Kakiuchi, S.; Aono, Y.; Huang, J.; Sato, S., et al. A novel targeting therapy of malignant mesothelioma using anti-podoplanin antibody. J Immunol 2013, 190, 6239-6249, doi:10.4049/jimmunol.1300448.

  1. Did the authors check if their used tumor cells line expresses ectopically Fc receptors, even at low level, but enough to show difference in “affinity” (as demonstrated in many human tumors) which would have been make the difference since mouse/rat does not bind to human Fc?

Reply: Unfortunately, we didn't. The protein dose is 40 micrograms. Even if the cells express ectopically Fc receptors, the effect would be small. We will consider Fc receptor expression in further studies.

  1. While preliminary data using subcutaneously (s.c.) tumor may be accepted, due to the unique natural history of malignant mesothelioma, the s.c. model is not appropriate to demonstrate toxicity and, much more important, therapeutically efficacy.

Reply: We agree with the reviewer's comment, and we are developing luciferase-expressing H226 cells to evaluate the efficacy in an orthotopic model. After establishing the cells, we will do that.

  1. No information regarding the immunogenicity of the chimera antibody are provide (due to the foreign parts, “chimera” part and the paratope of the chimeric antibody).

Reply: That is an important issue. The dose of NZ-16 is planning to be less than 4 mg per patient. This dose is very low for humans, suggesting a low risk to stop therapy due to the immunogenicity of NZ-16. In clinical RIT, the chimeric antibody Zevalin is widely used, and there is a small population dropping out due to the immunogenicity. The dose is similar to the dose of NZ-16 we plan. However, as the reviewer suggests, we have to consider immunogenicity, and we will carefully monitor patients who received radiolabeled NZ-16.

  1. No studies are shown with isotype antibody labeled to the alpha-emitters and 225Ac-lebeled NZ-12 antibody.

Reply: We already evaluated in vitro and in vivo properties of an isotype antibody in a previous study (reference below). Therefore, the present study includes no information about that.

Ref: Sudo, H.; Tsuji, A.B.; Sugyo, A.; Saga, T.; Kaneko, M.K.; Kato, Y.; Higashi, T. Therapeutic efficacy evaluation of radioimmunotherapy with 90Y-labeled anti-podoplanin antibody NZ-12 for mesothelioma. Cancer Sci 2019, 110, 1653-1664, doi:10.1111/cas.13979.

  1. No information regarding mycoplasma assessing and authentication of cells are provided.

Reply: We provided the certificate of analysis for the cells obtained from ATCC as supplementary information. That indicates mycoplasma negative.

Sincerely yours,

Atsushi B. Tsuji

Reviewer 3 Report

The Authors reported on the activity of radio-receptorial therapy, specifically alpha-radioimmunotherapy with the anti-Podoplanin antibody NZ-16, in mesothelioma cell lines and tumor-bearing mice. Authors evaluated the in vitro properties of radiolabeled antibodies via cell binding and competitive inhibition assays. Biodistribution of 111In-labeled antibodies was also investigated. These data were compared with those of a previous antibody, named NZ-12.

The topic is of interest, the manuscript is clear and well-written, methodology robust and the overall feeling is that of scientifically sound paper.

Author Response

Dear Reviewer 3,

The topic is of interest, the manuscript is clear and well-written, methodology robust and the overall feeling is that of scientifically sound paper.

Reply: We sincerely appreciate the reviewer for acknowledging the importance of our work. That strongly encourages us to conduct further studies.

Sincerely yours,

Atsushi B Tsuji

Round 2

Reviewer 1 Report

I thank the authors for their good answers to all my requirements and suggestions as a reviewer of this manuscript.

Reviewer 2 Report

Re: re-submission:

“Preclinical evaluation of podoplanin-targeted alpha-radioimmunotherapy with the novel antibody NZ-16 for malignant mesothelioma” - Cells

I would like to thank the authors for their comprehensive reviewing of the manuscript. The revised manuscript is trying to be responsive to previous reviewer comments. However, in their response and updated manuscript, the authors either avoided answering precisely to some of the critical questions or many raised topics were considered as limitations. They provided some answers, and not the pertinent answers. I am in doubt whether I have been able to communicate my questions properly to the authors.

For example, my question 9 did not ask for the original certificate released by ATCC. One of the major issue in research are cross-contamination of cell lines (using multiple cells lines in the lab) and contamination with mycoplasma during performing experiments. I was asking if the cells during performing the project were tested for mycoplasma and checked for authenticity periodically.

While I agree that the authors acknowledge majority of the issues raised by me as limitations, there is no way to move up for publication in a peer-review journal of repute avoiding these critical experiments to prove their hypothesis. Identification of the role of constant region in changing the affinity of scFv (very unusual), performing orthotopical model, and using second cell line (by knocking-in) are required for a pertinent scientifically sound research.

Finally, as minor comments, there is no discussion in all manuscript about the potential interaction of their antibody with normal PDPN-expressing cells. I also noticed that the authors mention that H226 cells were subcutaneously inoculated into male nude mice (BALB/c-nu/nu, 4 weeks old, CLEA Japan, Tokyo, Japan) (line 152). I do not recall any study using mice of 4 weeks of age. Is this correct?

When addressing rigor, transparency, and biological variables, both female and male should be investigated especially when the tumor model is not hormonal-dependent, as mesothelioma.